# Theoretical Investigation of the Effects of Aldehyde Substitution with Pyran Groups in D-π-A Dye on Performance of DSSCs

**DOI:** 10.3390/molecules29174175

**Published:** 2024-09-03

**Authors:** Suzan K. Alghamdi, Abdulaziz I. Aljameel, Rageh K. Hussein, Khalled Al-heuseen, Mamduh J. Aljaafreh, Dina Ezzat

**Affiliations:** 1Physics Department, Faculty of Science, Taibah University, Madinah 44256, Saudi Arabia; skghamdi@taibahu.edu.sa; 2Department of Physics, College of Science, Imam Mohammad Ibn Saud Islamic University (IMSIU), Riyadh 11623, Saudi Arabia; aialjameel@imamu.edu.sa (A.I.A.); maljaafreh@imamu.edu.sa (M.J.A.); 3Department of Applied Science, Ajloun University College, Al-Balqa Applied University, Ajloun 26873, Jordan; kalhussen@bau.edu.jo; 4Basic Science Department, Obour Institute (OI), Qalyubia 11828, Egypt; dinae@oi.edu.eg

**Keywords:** DFT, TD-DFT, D-π-aldehyde, pyran groups, UV–vis absorption spectra, monte carlo simulation

## Abstract

This work investigated the substitution of the aldehyde with a pyran functional group in D-π-aldehyde dye to improve cell performance. This strategy was suggested by recent work that synthesized D-π-aldehyde dye, which achieved a maximum absorption wavelength that was only slightly off the threshold for an ideal sensitizer. Therefore, DFT and TD-DFT were used to investigate the effect of different pyran substituents to replace the aldehyde group. The pyran groups reduced the dye energy gap better than other known anchoring groups. The proposed dyes showed facile intermolecular charge transfer through the localization of HOMO and LUMO orbitals on the donor and acceptor parts, which promoted orbital overlap with the TiO_2_ surface. The studied dyes have HOMO and LOMO energy levels that could regenerate electrons from redox potential electrodes and inject electrons into the TiO_2_ conduction band. The lone pairs of oxygen atoms in pyran components act as nucleophile centers, facilitating adsorption on the TiO_2_ surface through their electrophile atoms. Pyrans increased the efficacy of dye sensitizers by extending their absorbance range and causing the maximum peak to redshift deeper into the visible region. The effects of the pyran groups on photovoltaic properties such as light harvesting efficiency (LHE), free energy change of electron injection, and dye regeneration were investigated and discussed. The adsorption behaviors of the proposed dyes on the TiO_2_ (1 1 0) surface were investigated by means of Monte Carlo simulations. The calculated adsorption energies indicates that pyran fragments, compared to the aldehyde in the main dye, had a greater ability to induce the adsorption onto the TiO_2_ substrate.

## 1. Introduction

Small organic molecules are becoming more common than polymers as components of the basic building blocks of organic solar cells. This is due to the many benefits they provide, including their flexible electronic capabilities, high purity, and improved device reproducibility [1,2,3]. Even conventional inorganic materials have been replaced by organic alternatives due to the convenience of controlling the physical properties of these materials and their broad molecular design [4,5]. Therefore, among the most promising developments for converting solar energy are dye-sensitized solar cells (DSSCs), which are based on organic dyes. In general, organic sensitizers for DSSCs have donor–acceptor structures of the D-π-A type. The donor (D) and acceptor (A) fragments found in the structures of organic dyes can be altered using various components to satisfy the ideal photoelectrical parameters [6,7]. The exceptional performance of TiO_2_ in DSSCs has led to its designation as the state-of-the-art photoanode material. Remarkably, the morphological pattern, crystallographic phase, and surface characteristics are among the factors that contribute to the superior performance of TiO_2_. Comprehensive studies that address how π spacers, of different types and positions, affect the fundamental characteristics of DSSCs have been performed. Many popular π-spacers, including phenyl, polyene, thiophene, benzene derivative, and furan ring π-spacers, have been explored to improve the light-harvesting efficiency and enhance the functionality of DSSC systems [8,9,10,11,12]. The donor parts employed in organic DSSCs must be non-planar, redox stable, and electron-rich to minimize intermolecular dye aggregation [13,14]. The most common donor moiety, triphenylamine (TPA), possesses structural diversity that offers a range of organic dye geometries, which can lead to major differences in optoelectronic properties [15,16,17]. The strong tiedown of the acceptor fragment on the semiconductor surface is correlated with the withdrawing strength of the acceptor group, which could give rise to different HOMO and LUMO energy level positions and, in turn, distinct driving forces for electron injection and photocurrent density tuning [18,19,20]. The development of better DSSCs depends on the ability to comprehend the structure of the various acceptor groups, also known as anchor groups, to explore new ones. Recently, the chemical and physical properties of a wide range of acceptor groups have been studied to find desirable interactions at the dye–metal oxide interface [21,22,23,24]. The most effective anchors for modifying the properties of organic dyes in DSSCs are the carboxylic acid and cyanoacrylic acid groups since they provide the maximum redshift of the absorption spectra and the lowest energy gap [25,26]. Phenol, phosphonate, rhodanine-3-acetic acid, and many other anchoring moieties have all been studied both theoretically and experimentally [27,28,29].

Pyran is a ring made up of five carbon atoms and an oxygen atom, constituting a heterocyclic compound. Pyrans have been known to be extremely active compounds since their discovery in 1962, when they were obtained through the pyrolysis of 2-acetoxy-3,4-dihydro-2H-pyran [30,31]. There are extensive applications for pyrans in pigments, agrochemicals, and pharmaceutical domains where they exhibit anticancer, anti-allergic, and antibacterial activities [32,33,34,35]. Pyrans are also utilized as important components in solar cells and sensors and as prospective photo redox catalysts [36,37,38]. In the design of DSSCs, pyran components were frequently used as a core that functionalized with electron acceptor groups and were implemented at the donor parts in most investigations [39,40,41]. Within the limits of the research that has been performed, pyran compounds were very rarely selected to be used as acceptor components. 

The use of density functional theory (DFT) and time-dependent DFT (TD-DFT) quantum chemical calculations has grown considerably in recent years in molecular-level material investigations. They represent an extensive library of computational techniques for accurately calculating the electronic and optical properties of molecular structures with minimal computation costs [42,43]. DFT and TD-DFT played a major role in the design, analysis, and development of organic dyes for use in solar cell applications. For a large array of materials used in DSSCs, DFT and TD-DFT have yielded impressive results in the study and tuning of excited state properties, band gaps, and intermolecular charge transfers [44,45,46,47]. 

In this work, an acceptor aldehyde of a D-π-A dye was replaced with pyran derivatives to improve its electronic and spectroscopic properties. DFT and TD-DFT investigations of the newly designed dyes were conducted to highlight the modifications in the electronic structures and absorption spectra induced by the pyran components. The efficiency of the DSSCs using the proposed dyes was then assessed by analyzing their band gap values, UV–visible spectra, photovoltaic properties, and the energy levels of the frontier molecular orbitals.

## 2. Chemical Design

One popular technique for synthesizing D-π–A dyes is the Knoevenagel reaction, which involved the reaction of D–π– aldehyde with cyanoacrylic acid. Furthermore, Suzuki cross-couplings are frequently used to build molecules of the D-π–aldehyde type, making the synthesis of aldehydes a key process in the manufacture of sensitizers. Recently, a research team led by Nikita S. Gudim et al. prepared a D-π–A dye, which has a thiophene spacer group coupled to an aldehyde moiety, via the Knoevenagel reaction [48]. The new synthetic compound 5-(9-(p-tolyl)-2,3,4,4a,9,9a-hexahydro-1H-1,4-methanocarbazol-6-yl) thiophene-2-carbaldehyde (Figure 1) was prepared in that work as a precursor for the development of DSSC components. The study also included spectroscopic analyses to determine the structure of the synthesized compound, such as IR spectroscopy, UV–vis spectroscopy, and high–resolution mass spectrometry. The UV–vis absorption spectra obtained in a solution of DCM showed two absorption maxima, with the greatest wave absorption maximum reported at 432 nm (Appendix A). This value is slightly outside the visible to near-infrared range that an effective organic sensitizer should have. The authors of that work therefore clearly implied that another functionalization at the aldehyde group would produce a redshift band that could eventually generate a D-π-A-type dye with a broad spectral range. Drawing from this, the compound was redesigned by substituting the aldehyde group with heterocyclic pyran groups to provide competitive electronic and absorption capabilities. As illustrated in Figure 1, three D-π-A dyes were derived from the basic compound with three different pyran moieties in place of the aldehyde acceptor group. The basic compound will be abbreviated as BC, and the three designed compounds will be referred to as BC-Pyran1, BC-Pyran2, and BC-Pyran3.

## 3. Results

An essential factor to consider when determining a dye’s effectiveness for usage in DSSCs is its energy gap ($E_{g}$). $E_{g}$ can be defined as the difference in energy between the highest occupied molecular orbital (HOMO) and the lowest unoccupied molecular orbital (LUMO). Dyes with smaller energy gaps can absorb longer wavelengths of light more readily, which enhances the overall efficiency of the DSSCs by expanding the spectrum of solar energy that can be captured by the device. Therefore, reducing a dye’s energy gap is one of the key strategies to increase the efficiency of organic solar cells. Among the chemical reactivity parameters that can give insight into a dye’s chemical nature are ionization potential (IP) and electron affinity (EA). IP describes the energy needed for removing an electron from an atom (IP = −E_HOMO_), whereas electron affinity measures the energy released when an electron is added to an atom (EA = −E_LUMO_). The full-electron donor–acceptor map (FEDAM) by Martinez states that dyes with a high EA and low IP have good donor and acceptor properties [49]. The HOMO and LUMO energies, as well as $E_{g}$, IP, and EA, of the studied dyes were calculated and are summarized in Table 1. The calculated energy gap recoded for the base compound (BC) was 3.37 eV. The strong cyanoacrylic anchoring group that has been proven to significantly decrease the energy gap of other investigated DSSCs was also tested. The cyanoacrylic acid group was able to reduce the energy gap to 2.73 eV. The chosen pyran groups were able to achieve an energy gap that was smaller than that of the BC. The pyran 2 moiety outperformed the strong cyanoacrylic group, recording a lower energy gap (2.61 eV). Thus, pyran groups could be the focus of further research since they can affect the DSSC energy gap more than other known anchoring groups. The BC-Pyran2 dye had the lowest IP and an average increase in its EA value, as shown in Table 1. Therefore, it is expected to have the easiest electron flux since it has good electron donor and acceptor properties.

Intramolecular charge transfer (ICT) within a molecule can be determined by analyzing the electron density distribution throughout the frontier molecular orbitals (FMO) over its surface. HOMO and LUMO distributions on molecule surfaces could determine whether ICT is easy. The required proper distribution for easy ICT is satisfied if the HOMO is distributed over the donor parts and the LUMO is distributed over the acceptor parts; different distributions could make ICT more difficult. Figure 2 displays the HOMO and LUMO electron density distributions for the studied dyes. The HOMO were bonding orbitals that were dispersed throughout the p-tolyl and hexahydro-1H-1,4-methanol carbazole fragments of the donor component, except for its upper section. Furthermore, bonding orbitals built on the thiophene ring constituted a portion of the HOMO. The LUMO was formed by non-bonding orbitals that accumulated on the thiophene spacer and the acceptor moieties. Hence, an effective dye-sensitizer that exhibits facile ICT was proven by the localization of the HOMO and LUMO on the donor and acceptor units, respectively. Furthermore, the integration between the HOMO and LUMO, as shown in Figure 2, had a role in promoting ICT and electron injection, which in turn led to improved orbital overlap with the TiO_2_ conduction band (CB). 

It is essential to have knowledge of the energy levels of the dyes to be able to properly explain the electron injection and dye regeneration processes in DSSCs. Optimizing the efficiency of DSSC devices involves important considerations such as keeping the HOMO energy level lower than the electrolyte redox potential I−/I3− (−4.8 eV) and the LUMO energy level above the TiO_2_ CB (−4.0 eV) [50]. More electron injection efficiency results from a higher E_LUMO_ relative to the lower edge of the semiconductor CB. The process of electron regeneration from the TiO_2_ CB to the redox potential is facilitated by lowering E_HOMO_ below the redox energy level. Figure 3 shows the energy level diagrams of the proposed dyes, the electrolyte redox potential, and the CB of TiO_2_. Electrons can be successfully injected from the excited state of the dyes into the CB TiO_2_, since the LUMO energy levels of the studied dyes are above the CB edge of TiO_2_. It is also noted that the E_HOMO_ of the dyes is lower than the redox energy level of the electrolyte, which promotes the back-electron transition from the CB of TiO_2_. The effect of the pyran moieties on declining LUMO levels and rising HOMO levels relative to BC is clearly seen in the figure. This effect will be considered in the evaluation of the photovoltaic parameters.

A technique known as “Molecular Electrostatic Potential” (MEP) displays the electrostatic potential of reactive sites in molecules based on the three-dimensional electrostatic contribution of the charge distribution. MEP is defined as the predicted value of the reciprocal vector operator in the context of quantum mechanics, which is the electrostatic interaction energy between the positive unit charge and the molecule charge distribution [51]. The electrostatic potential from the MEP analysis is shown in blue, red, and white on a map. A positive electrostatic potential or an electron-deficient region is indicated as a blue region, a negative electrostatic potential or an electron-rich area is indicated as a red region, and a zero potential region is indicated by white. The regions with the strongest attraction and are the most susceptible to nucleophilic attack are the blue regions, while the regions with the strongest repulsion and are the most susceptible to electrophilic attack are the red spots. The optimized structures were used to calculate the MEP for the proposed dyes in their ground state, as shown in Figure 4. Positive potential spots were mainly seen above the hydrogen atoms for both the thiophene ring and the acceptor moieties (aldehyde or pyran groups). These sites served as indicators for potential locations of nucleophilic attack. Negative potential, where an electrophilic attack can potentially occur, was found with a greater extent on the oxygen atoms of the acceptor components and barely on the nitrogen atom of the donor part. More negative spots (red color) were seen in the pyran components due to the higher number of lone pairs of oxygen atoms than in the aldehyde group. The side face of BC-Pyran2 (the side that absorbs onto the TiO_2_ surface) exhibited a large negative potential, but this potential was lower in BC-Pyran1 and BC-Pyran3 due to the presence of an adjacent hydrogen atom. The adsorption process on the metal surface, whose atoms operate as electrophiles, was greatly aided by the free electron pairs that act as nucleophile centers on the acceptor group and are easily available for sharing to form a bond [52]. From this perspective, it is expected that BC-Pyran2 has the greatest potential to be adsorbed onto the surface of TiO_2_.

The TD-DFT method was used to calculate the light-absorbing properties of the studied dyes in dichloromethane. The solvent model SMD was employed at the MPW1PW91/6-311G** theoretical level. Figure 5 displays the UV–vis spectra, whereas Table 2 lists the electronic excited states and related absorption properties. As shown by the figure, each dye exhibited two bands of absorption: a smaller band in the region between 300 and 400 nm and a broader, more intense band in the visible region. The maximum absorption was the ICT band responsible for the charge transfer from the donor to the acceptor, while the smaller band was associated with the π-π* electronic transition. The pyran moieties expanded the spectral range and resulted in a redshift in the absorption bands. BC-Pyran2 produced the largest redshift, which was 100 nm relative to the basic compound. The redshifted absorption peaks and their broadness improved the efficiency of the organic sensitizer. The excitation from the HOMO to the LUMO was the dominant contribution to the main absorption band, as indicated by Table 2. The oscillator strength increases observed in BC-Pyran1 and BC-Pyran2 suggested that these dyes exhibited the most potential change in the excitation process.

The overall conversion efficiency (*η*), which represents a performance metric for DSSCs, is determined by two main factors [53]: (1)$$\eta \propto J_{SC}\;*\;V_{OC}$$
where *J_SC_* is the short-circuit photocurrent density and $V_{OC}$ is the open-circuit voltage. The short circuit current $J_{SC}$ is determined by the dye’s absorption coefficient as well as the interaction between the dye and the TiO_2_ surface. It follows that one of the fundamental factors in determining *J_SC_* is the light-harvesting efficiency (LHE). LHE provides a scale for photon absorption and the efficiency of excited electron injection and can be calculated as follows [53]:(2)$$LHE=(1-10^{-f})$$
where ƒ is the oscillator strength associated with the absorption wavelength of the excited state. One of the key photovoltaic parameters that determines the efficiency of DSSCs is the open-circuit voltage ($V_{OC}$). The $V_{OC}$ is primarily determined by the values of *E_LUMO_* and the energy of the TiO_2_ CB according to the equation [54]:(3)$$V_{OC}=E_{LUMO}-E_{CB}^{TiO2}$$

The energy value of the TiO_2_ CB $\left(E_{CB}^{TiO2}\right)$, as stated before, is −4.0 eV. The electron injection rate and, in turn, the *J_SC_* in DSSCs are affected by the free energy change for electron injection $({\Delta G}^{inject}$), which can be expressed as [55]
(4)$${\Delta G}^{inject}=E_{ox}^{dye*}-\left|E_{CB}^{TiO2}\right|$$$E_{ox}^{dye*}$ represents the dye’s oxidation potential in the excited state and is denoted as [55]
(5)$$E_{ox}^{dye*}=E_{ox}^{dye}-E_{ex}$$$E_{ox}^{dye}$ refers to the redox potential of the ground state (−*E_HOMO_*), whereas $E_{ex}$ is the lowest-energy electronic transition associated with the maximum absorption wavelength in the UV–visible absorption spectrum. There must be a strong driving force for the regeneration of the dye by the redox electrolyte. This could be clarified with the dye regeneration energy ${\Delta G}^{reg}$, which is defined as [55]
(6)$${\Delta G}^{reg}=E_{ox}^{dye}-\left|E_{redox}^{electrolyte}\right|$$

As stated above, an electrolyte redox $E_{redox}^{electrolyte}$ value of the redox couple iodide/triiodide of −4.80 eV is routinely utilized. The dye’s charge transfer properties are strongly affected by its excited-state lifetime (τ). A higher charge transfer efficiency would be provided by a dye with a longer excited state lifetime (calculated using Equation (7)) [55].
(7)$$\tau =\frac{1.499}{ƒE_{ex}^{2}}$$

Table 3 provides the most important photovoltaic parameters for the proposed dyes. There is sufficient driving force for the electron injection and regeneration processes since the calculated ${\Delta G}^{inject}$ and ${\Delta G}^{reg}$ values of all the dyes are higher than 0.2 eV [56,57]. An increase in the driving force for electrons injected into the TiO_2_ CB from the excited states of the dyes correlated with a more negative value of ${\Delta G}^{inject}$. Considering two dyes with similar structures, the dye with the higher excited state relative to the TiO_2_ edge would have a more efficient electron injection and open-circuit voltage [58,59]. This is obviously reflected in the calculated values of ${\Delta G}^{inject}$ and $V_{OC}$ in Table 3. The negativity of ${\Delta G}^{inject}$ and the values of $V_{OC}$ decreased because of the substituent pyran groups, since pyran groups lowered the E_LUMO_ in BC. A high value of regeneration energy denotes that the dye has the potential to increase the photoelectric conversion efficiency by recovering electrons from the electrolyte [60]. The modified dyes could exhibit a lower yield of electron collection and dye regeneration due to having lower regeneration energy values than the BC. All the dyes were expected to be potential photosensitizers, given that the calculated LHE values varied between 85 and 87%. The excited state lifetime increased in the following order: BC-Pyran3 > BC-Pyran2 > BC-Pyran1 > BC. These results imply that pyran moieties increase the dye’s excited state lifetime, suggesting that the cationic state for the studied dyes is the most optimal state for effective charge transfer.

Organic dye adsorption onto a metal oxide surface is commonly simulated to obtain insight into the nature and strength of the interaction of the dye molecules with a particular crystal facet of the metal oxide substrate. It is possible to precisely determine the most stable configuration of the organic dye when it is adsorbed onto a metal surface by evaluating the adsorption energy, rigid adsorption energy, and deformation energy. The adsorption energy represents the energy necessary for the relaxed adsorbed components to be adsorbed onto the substrate’s surface. The energy released or needed when unrelaxed adsorbates (before geometry optimization) adsorb onto the substrate is measured by the rigid adsorption energy. The deformation energy represents the energy released when the adsorbate relaxes because of the geometry optimization on the substrate surface. Figure 6 displays the side views of the most stable adsorption configurations for the dyes BC, BC-Pyran1, BC-Pyran2, and BC-Pyran3 on the TiO_2_ (110) surface. The adsorption energies for relaxed adsorbate dyes, the deformation energies for relaxed adsorbate dyes, and the rigid adsorption energies for unrelaxed adsorbate dyes are all listed in Table 4. The total energy of the substrate/adsorbate arrangement, which is the sum of the individual energy components, is also included in the table. $\frac{dE_{ad}}{d\;Ni}$ provides the energies of the adsorbate molecules after removing one component of the adsorbate.

The BC-Pyran2 and BC-Pyran3 dyes were adsorbed in a nearly planar arrangement on the TiO_2_ (110) surface, as shown by the contact angles between them and the surface in Figure 6, which were approximately 180°. This implies an improvement in the adsorbate–adsorbent contact surface. The adsorption energy values for the studied adsorption configurations were negative, as illustrated in Table 4, indicating a spontaneous and an exothermic reaction. A large absolute value of the adsorption energy indicates strong adsorption activity. A higher negative value of $\frac{dE_{ad}}{d\;Ni}$ for the adsorbate dyes than that of the DCM solvent indicates that the organic dyes have a higher affinity for adsorption onto the TiO_2_ surface than the solvent. The system BC-Pyran2 was the most stable and exhibited the strongest adsorption, as evidenced by the large negative value for the calculated adsorption energy in Table 4. BC-Pyran1 and BC-Pyran2 had recorded adsorption energy values of −6399.78 and −6365.01 Kcal/mol, respectively. These values are much higher than the value of BC (−4173.05 Kcal/mol), revealing that the pyrans increase the ability of the dyes to bind to the TiO_2_ surface compared to the primary dye.

## 4. Computational Methods

The ground state optimization for the designed sensitizers was performed using the Gaussian 09 software program [61]. Time-dependent density functional theory (TD-DFT) was utilized to calculate the UV-vis absorption spectrum along with associated optical parameters, including oscillator strength (f), excitation energy, and the contribution of the molecular orbitals in the electronic transitions. The solvent effect was included using the universal continuum solvation model (SMD), with dichloromethane (DCM) as the solvent. To find the most accurate functional for the calculations, several well-known DFT formalisms were considered. First, the ground-state structure was predicted using several functionals with the standard basis set 6-311G**. The maximum absorption wavelength was then determined using the TD-DFT approach and compared to the experimental findings in the literature (Appendix A). The MPW1PW91 function was selected among the tested ones since it yielded the maximum wavelength (435 nm) that was most like the experimental value (432 nm). The proposed dyes were designed and visualized using the molecular editor and visualization Avogadro tool, version 1.2.0 [62].

The tool used for the adsorption calculations to simulate the adsorbate dyes on the TiO_2_ substrate was the Adsorption Locator module included in the Biovia Materials Studio software 2017 (MS2017) [63]. The Adsorption Locator module uses Monte Carlo simulations of the adsorbate–substrate configuration to identify potential adsorption sites as the temperature gradually decreases during the molecular dynamics modeling of the simulated annealing procedure. The adsorption model, made up of three layers of TiO_2_ rutile (110), was used to construct a simulation box with dimensions of 26.63 Å x 51.97 Å x 24.03 Å. Then, a 15-Å vacuum slab along the Z axis was added. The geometry optimization of the studied dyes was carried out using the COMPASS force field in the Forcite module. One dye molecule and one hundred and fifty molecules of the DCM solvent were introduced into the vacuum box in each simulation for the annealing calculations.

## 5. Conclusions

The current research studied the effects of substituting the aldehyde group in D-π-A dye with pyran moieties using DFT and TD-DFT methods. The pyran moieties were able to reduce the energy gap and give the dye favorable electron donor and acceptor properties. The proper arrangements of the HOMO and LUMO on the donor–acceptor components of the proposed dyes provide a flexible ICT. The MEP revealed nucleophile sites in the pyran components, characterized by lone pairs of oxygen atoms, which facilitate their binding to the TiO_2_ surface. The molecular orbital energies demonstrated the thermodynamic feasibility of electron injection into the TiO_2_ CB and dye regeneration through redox potential. Based on the calculated UV-vis absorption spectra, the dyes with pyrans exhibited more strong and broader absorption bands in the visible spectrum compared to the initial dye. The pyran dyes demonstrated considerable light-harvesting efficiency in their photovoltaic properties, as indicated by a larger oscillator strength, a higher LHE, and a longer radiative lifetime. In the study of the dyes’ adsorption onto a TiO_2_ (110) surface, BC-Pyran2 attained the largest adsorption energy (−6399.78 Kcal/mol) relative to the primary dye BC (−2063.38 Kcal/mol). The presence of different pyran substituents in the acceptor part had a stronger effect on the DSSC’s performance, which encourages the design of D-π-A with pyrans as acceptor moieties in the future construction of more efficient DSSC sensitizers.

## Figures and Tables

**Figure 1 molecules-29-04175-f001:**
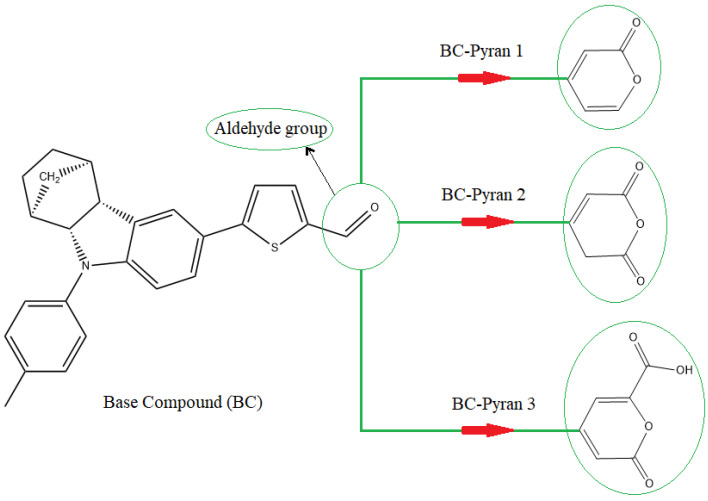
The construction of proposed dyes by substituting the aldehyde group in the base compound with three different pyran moieties.

**Figure 2 molecules-29-04175-f002:**
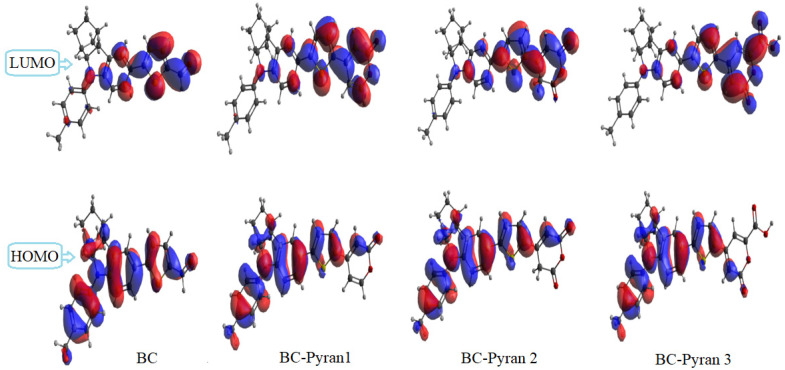
FMO contours of the HOMOs and LUMOs of the studied dyes.

**Figure 3 molecules-29-04175-f003:**
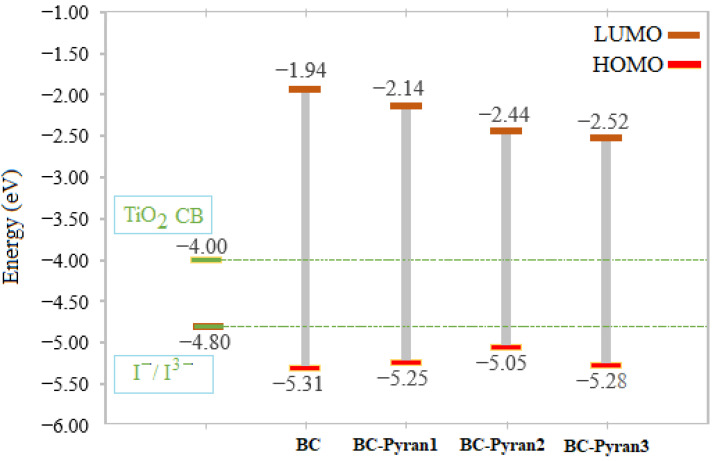
The HOMO and LUMO energy levels for the base compound and the designed dyes.

**Figure 4 molecules-29-04175-f004:**
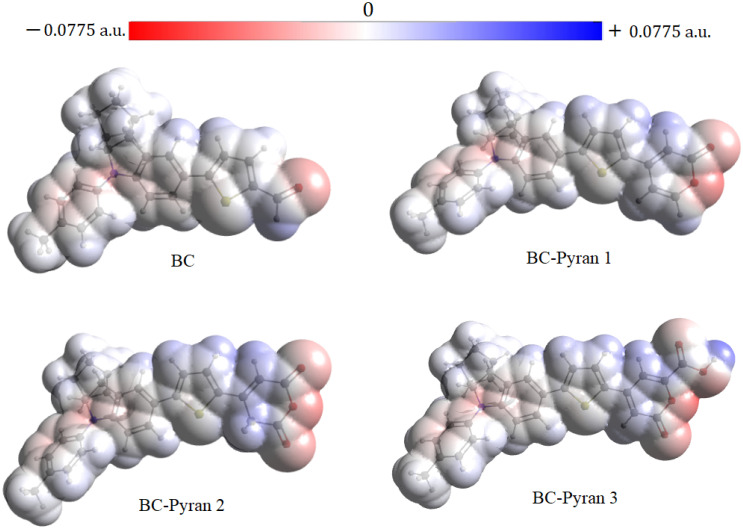
The 3D map of the molecular electrostatic potential of BC and the designed dyes.

**Figure 5 molecules-29-04175-f005:**
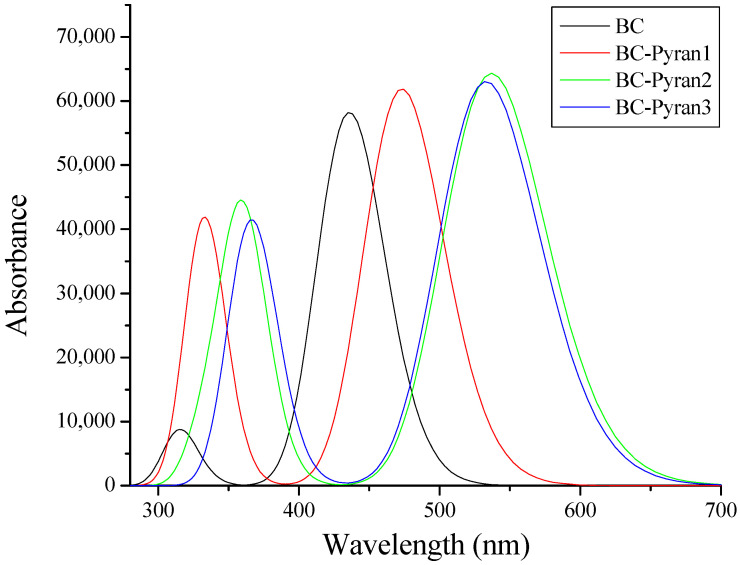
TD-DFT simulation of the absorption spectra of the studied dyes at the MPW1PW91/6-311G** level in dichloromethane.

**Figure 6 molecules-29-04175-f006:**
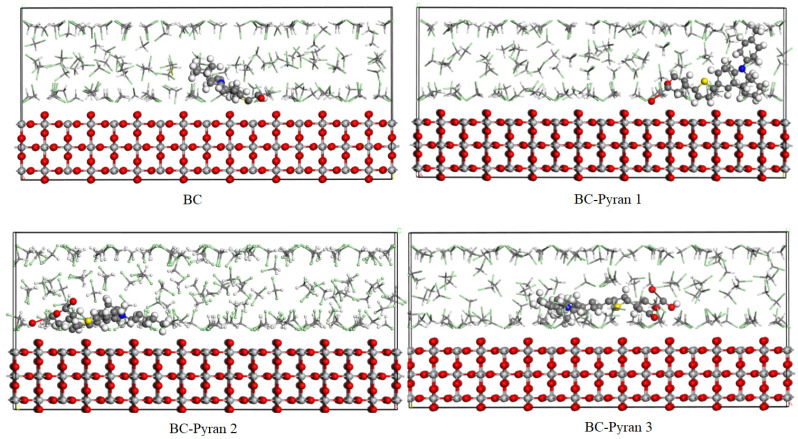
Side views that show stable adsorption configurations of the BC and BC-pyran dye/TiO_2_ (110) systems.

**Table 1 molecules-29-04175-t001:** The calculated values of HOMO and LUMO energies, energy gap ($E_{g}$), ionization potential (IP), and electron affinity (EA) for the studied dyes.

Compound	E_HOMO_ (eV)	E_LUMO_ (eV)	$E_{g}$ (eV)	IP (eV)	EA (eV)
BC	−5.31	−1.94	3.37	5.31	1.94
BC-Pyran1	−5.25	−2.14	3.11	5.25	2.14
BC-Pyran2	−5.05	−2.44	2.61	5.05	2.44
BC-Pyran3	−5.28	−2.52	2.76	5.28	2.52
BC + cyanoacrylic group	−5.34	−2.61	2.73	5.34	2.61

**Table 2 molecules-29-04175-t002:** Excitation energy ($E_{ex}$), absorption wavelength, oscillator strength (ƒ), and the major contribution of the electronic transitions for the studied dyes.

Compound	$E_{ex}$ (eV)	Wavelength (nm)	Oscillator Strength (ƒ)	Transition	Major Contribution
BC	2.85	435.49	0.81	HOMO→LUMO	99%
BC-Pyran1	2.63	472.82	0.85	HOMO→LUMO	99%
BC-Pyran2	2.32	536.66	0.89	HOMO→LUMO	99%
BC-Pyran3	2.33	532.696	0.87	HOMO→LUMO	99%

**Table 3 molecules-29-04175-t003:** The calculated photovoltaic descriptor parameters of the studied dyes.

Compound	$E_{ex}$ (eV)	$E_{ox}^{dye}$ (eV)	$E_{ox}^{dye*}$ (eV)	${\Delta G}^{inject}$ (eV)	${\Delta G}^{reg}$ (eV)	LHE	$V_{OC}$ (eV)	τ (ns)
BC	2.85	5.31	2.46	−1.54	0.51	0.85	2.06	0.23
BC-Pyran1	2.63	5.25	2.63	−1.38	0.45	0.86	1.86	0.25
BC-Pyran2	2.32	5.05	2.74	−1.27	0.25	0.87	1.56	0.31
BC-Pyran3	2.33	5.28	2.95	−1.05	0.48	0.87	1.48	0.32
Best candidate	BC-Pyran2	-	-	BC	BC	BC-Pyran2 and BC-Pyran3	BC	BC-Pyran2

**Table 4 molecules-29-04175-t004:** The MD simulation results for the adsorption of the specified dyes onto the TiO_2_ (110) surface.

	Total Energy (Kcal/mol)	Adsorption Energy (Kcal/mol)	Rigid Adsorption Energy (Kcal/mol)	Deformation Energy (Kcal/mol)	Dye: $\frac{dE_{ad}}{dNi}$	DCM: $\frac{dE_{ad}}{dNi}$
BC	−2063.38	−4173.05	−2135.33	−2037.72	−2100.65	−8.63
BC-Pyran1	−2068.47	−4363.10	−2138.70	−2224. 40	−2289.77	−10.18
BC-Pyran2	−2063.26	−6399.78	−2132.81	−4266.97	−4325.64	−8.92
BC-Pyran3	−2076.47	−6365.01	−2161.74	−4203.27	−4258.79	−9.92

## Data Availability

The original contributions presented in the study are included in the article; further inquiries can be directed to the corresponding author.

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
