# Peer review of "Theoretical Investigation of the Effects of Aldehyde Substitution with Pyran Groups in D-π-A Dye on Performance of DSSCs"

_molecules, 2024, doi:10.3390/molecules29174175_

Round 1

Reviewer 1 Report

Comments and Suggestions for Authors

PEER REVIEW REPORT

The paper titled “Theoretical Investigation of the Aldehyde Group Substitution

Effect in D-π-A Dye for a Competitive Performance of DSSCs” investigated the use of Pyran-substituted dye molecule when incorporated as acceptor moieties in the 

dye-sensitized solar cells (DSSCs). The article presents theoretical investigations on molecular electronic structure, vibronic absorption spectrum, photovoltaic properties, and the adsorption sites to a TiO2 surface of 3 proposed Pyran-substituted compounds. By comparing the predicted quantities with those of an aldehyde-group-based dye (base compound) that was experimentally demonstrated, the authors suggest the use of D-π-A structure with Pyrans as acceptor moieties for improved efficiency in DSSCs.

The authors present a computational chemistry study—based on density functional theory analysis and Monte Carlo simulations—and provide computational design insights into the anchoring group design in the dye-sensitized solar cells. Thus it matches the scope in the “Computational and Theoretical Insights on Molecular Structure, Solvation, Interactions and New Materials DesignSpecial Issue. I agree that the topic is of interest and of some significance to the community of computational chemistry and materials design. The Pyran-group-substituted chemical structures considered in the article were not seen in the literature before. 

However, with the above being said, I don’t think the article is ready for publication in its current state, as there is much room for improvement in terms of (i.) the narrative and presentation of the scientific question for better readability and (ii.) the scientific methods that are employed in the study for scientific soundness. The novelty and the scope of the article are quite narrow (for instance, compared to a previous study in a very similar context, Ref [45], https://doi.org/10.3390/ijms25137138). With some reworking, I envision the article could be much more engaging for the reader and the conclusion could be supported by the results much stronger. Overall, I would suggest a reconsideration after major revisions of the article to Molecules Editorial Office. Below, please find my detailed comments.

a) General Comments

[1] The results and conclusions are not convincing. There is much to discuss regarding the computational protocols that are employed in the study (see comment [4] below). Even if we assume the protocol is fine for now, a large missing piece is how those results can guide the design of the DSSC experiments. Only one experiment data point is presented throughout the article to validate the theoretical results, and in my opinion, it is far from enough. Moreover, in the energy level results, BC (cyanoacrylic group) still has the best EA, and the Eg and IP difference between BC-Pyran2 and BC could be under theoretical uncertainty (given that not enough benchmarks are provided in the article). It is not the case that a small “recorded high” energy value guarantees a successful design of the solar cells. The authors need to bridge the gap between the evaluated quantities and the realistic cell performance.

Another example is on Line 125, which reads that “Intramolecular charge transfer (ICT) within a molecule can be determined by analysis of the electron density distribution”. What is the degree of “determined” and what can be “determined”? The rate of charge transfer would certainly be a dynamical process and the accurate determination of it would highly likely involve statistical sampling of the reorganization energy from Boltzmann-distributed geometries; molecular dynamics simulation may be needed to correct for the dynamical recrossing factor and to devise a reaction mechanism if needed. The electron transfer mechanism could be different when different dyes are involved. A frontier molecular orbital analysis does not provide such information. The authors make strong claims but fail to provide evidence to support the claims.

The scope of the presented study is also narrow. It focuses on DSSC, then one specific realization of DSSC with TiO2 based architecture, and it tested 3 possible D-\pi-A dyes associated with a very specific functional group as an anchoring group. The whole article is presented in a very particular context. For readers of Molecules who are not necessarily an expert on DSSCs, the article needs to provide a higher-level scope to convince them the work has enough novelty and impact. For example, the authors could consider the following questions to answer: What about the other Pyran-based structures that are not included in the 3 structures included in the study? Do the results in the study suggest a systematic way to design the acceptor moieties, or let’s say, Pyran-based moieties for the DSSC? If we look at the currently designed dyes, they are not considerably better than the previous ones and the results may only have limited significance to the community. 

[2] Certain scientific fragments are missing or misordered in the presentation of the article. The lack of sufficient background information or computational details creates a significant barrier for the readers to understand the results. The computational details are not adequately described for the reproduction of the data in the current study.  Specifically, 

2.1 The article would benefit significantly if a paragraph to describe the reactive process (maybe also including the adsorption process on the surface) could be provided before the Results section. It helps to build the connection that why would design a dye structure where “[the orbitals] has a role in promoting ICT and electron injection, which in turn leads to improved orbital overlap with the TiO2 conduction band”, “[the dye] has the greatest potential to be adsorbed on the surface of TiO2”, or “[these dyes] exhibit the most potential change in the excitation process”. Only with a description of the chemical process, the readers understand what quantities are important in the design and those calculations become meaningful.

2.2 How is the absorption of Pyran-substituted compounds calculated in the work (Figure 4)? The article states that “The maximum absorption wavelength is then determined using the TD-DFT”, and it reads to me that a stick spectrum (i.e. peak position and intensity) can be obtained with TD-DFT calculations. Then how is the lineshape and peak width obtained in this study? Is a certain dynamics model for the nuclei assumed in the calculation to predict the absorption spectrum? For example, assuming a DHO model and for the size of the molecules considered here, those calculations are affordable and can provide much insight. Note that as the authors claimed, the peak positions and the broadness are crucial to DSSC performance. The computational details provided are not sufficient enough to understand the results. 

2.3 On the computational protocol, Line 324, what’s the reason that “6-311G**” is the standard basis? Have the authors investigated the effects when other basis sets are employed (e.g. double-zeta basis from def family?) “Several functionals” - what are those functionals? Can authors report the benchmark results?  Line 328, what are the “test ones”? Can authors also report the benchmark results? Fig.S1 is among the examples I am looking for, and can the authors provide more experimental quantity validation to show that this density functional captures the excitation behavior correctly, rather than a lucky error cancellation? It looks to me that the authors indeed benchmark their calculations against available experimental values for BC base compound, then use the same protocol for the prediction for Pyran 1, 2, 3 compounds. The content, if stated more clearly, will strengthen the validity of the simulation results. 

2.4 The protocols of adsorption calculation are ambiguous. Line 334 states that it is a Monte Carlo approach with the simulated annealing procedure, but why “during the molecular dynamics”? Is it a Molecular Dynamics simulation? Those two are completely different simulation protocols. Also the word “MD” appears in the caption of Table 4. Does the current calculation involve time-dependent trajectories? Can authors report those? 

The quantities that are reported in Table 4 could also be described more clearly. Line 258 states that the adsorption energy “represents the energy necessary for the relaxed adsorbed components to be adsorbed onto the substrate's surface”. Does it mean the energy of the geometries where the dye molecule is located at the optimal location on the TiO2(110) surface? Is the surface/lattice relaxation considered here or not? If not, how good is the frozen surface approximation for similar systems in the literature results or benchmark studies?

In addition, how is the value dE_ad/dN_i evaluated? Line 269 reads “dE_ad/dNi provides the energies of the adsorbate molecules after removing one component of the adsorbate.” What is the “one component of the adsorbate” to remove? I originally thought that the total adsorption (or desorption, since the thermodynamical cycle can be reversed) energy is the sum of adsorption energy plus the deformation energy but it does not match the numerical results. Can authors explain what’s their definition of dE_ad/dN_i? The authors also presented dE_ad/dN_i for DCM. (By the way, the units for the last two columns are missing in Table 5.) Does it mean the desorption of a DCM molecule on the TiO2(110) surface? Or does it mean the desorption energy of the dye molecule off the DCM bulk? The latter is not an adsorption process. The authors also need to clarify to avoid confusion among potential readers.

2.5 The chemical structure of the considered dyes (Figure 6 and associated texts) is presented AFTER the whole section of results. In my opinion, it creates unnecessary difficulty for the readers to understand the Results.

Line 292 to 307 provide fundamental and very helpful background knowledge (!) of the whole study. Moving this part in front of the Results section would help explain the motivation and goal of the functional group substitution in this study. Also, Reference [57] is a previous work that cannot be skipped and I suggest it be presented to the reader in the introduction given its relevance.

2.6 Line 136, where the TiO2 first appears in the Results section (“had a greater ability to induce the adsorption on the TiO2 substrate”, and note it also does not appear in the Introduction), is quite abrupt. The concept needs to be introduced first before a sudden mention of TiO2 conduction band energy.

[3] In several places where the authors review the literature contributions, I believe a brief but specific description of the literature discoveries helps the reader to build a better understanding of the context of the current work. What’s the current status of dye-sensitized solar cells? What’s the advantage of TiO2-film-based design? What about the other anchoring group of the cells? Are they promising or less promising, given a multi-dimensional metric of efficiency, stability, and charging/discharging behavior etc.? 

3.1 Instead of “Phenol, phosphonate, rhodanine-3-acetic acid, and many other anchoring moieties have all been studied both theoretically and experimentally” (Line 70), can authors summarize the performance of those anchoring groups (those not Pyran or cyanoacrylic acid)? 

3.2 On Line 80, it reads that “Within the limits of the research that has been done, Pyran compounds were very rarely selected to be used as acceptor components”, does the word “rarely selected” suggest that there exists, and not too many, previous work that considers the Pyran as the anchoring group? The authors will need to clarify and precisely state the status of literature, in particular since it is a major component of the novelty of the article.

3.3 On Line 88, “DFT and TD-DFT played a major role in the design, analysis, and development of organic dyes for use in solar cell applications.” Please implement citations and summarize the the status of literature studies. Reflecting “major role”, is DFT/TDDFT sufficient to capture the electron dynamical and static correlation in the context of DSSC design? Reflecting on “design, analysis, and development”, what properties did previous literature calculate with DFT/TDDFT for the design and analysis of DSSC performance? Especially, if this article calculates a property that was underestimated but indeed is crucial in DSSC design, this would greatly strengthen the novelty of the article. A similar argument can be made reflecting “impressive results” on Line 81? What are those impressive results? 

3.4 On Line 111, the reference [48] seems relevant to the DSSC design but is not mentioned in the literature review in the Introduction.

3.5 Please implement citations of the “Molecular Electrostatic Potential” methods that are used in the study. (Line 156) Also, a sentence to introduce how the electrostatic potentials are defined or calculated would be useful to the readers.

[4] I suggest an explicit mention of the Pyran functional group in the title as it is the major focus of the DFT study in the article. Also similarly in the abstract, “Pyran functional group” can be mentioned along with “the substitution of the aldehyde group” (Line 16) since the authors do not attempt the substitution of the aldehyde group by other possible functional groups as anchor groups. A clear and straightforward statement can be something like “The current research studied the effects of substituting the aldehyde group in D-π-A dye with Pyran moieties using DFT and TD-DFT methods” (Line 344).

[5] On Line 102, “Dyes with smaller energy gaps can absorb longer wavelengths of light more

readily”, what does the “more readily” mean here? Can the authors be quantitative, or at least more specific? And why is the case that “which enhances the overall efficiency of DSSCs by expanding the spectrum of solar energy that can be captured by the device”? If the point the authors wanted to make is that the longer wavelength absorption is more efficient than the short wavelength, it would be helpful to supplement a simple equation to show the scaling of efficiency v.s. wavelength.

[6] Table 1 and Figure 2 both report the energy levels from the calculations. Moreover, the data reported there have significant overlap. I would suggest a merge of the Table 1 and Figure 2, as the two results are currently separated by a discussion of intramolecular charge transfer in the Pyran compounds.

[7] On Line 192, can the authors explain why is the case that “The red-shifted absorption peaks and their broadness improve the ability to obtain an efficient organic sensitizer“? It is not straightforward to follow. Could the authors provide some argument in a quantitative manner?

[8] How was the “Major Contribution” in Table 2 calculated? Does it mean the contribution of HOMO->LUMO transition to the S0->S1 transition? Please clarify. 

[9] In the paragraph discussing the photovoltaic properties of the proposed dyes, multiple quantities are presented to provide a comprehensive metric to evaluate the conversion efficiency of the dyes in DSSCs. It would be helpful to present another row in Table 3, to straightforwardly rank the four dyes according to each matric. As an example, the additional row can be named “best candidate”, and the data can be “BC” for column E_ex, “Pyran2 for LHE”, and other columns of the same.

Presentation Comments

b) In my opinion, editions of the following words or sentences will help potential readers interpret the content with less ambiguity. Some confusions come from the use of the words, the others come from the organization and the flow of the journal article.

b1. “An uncommon instance of Pyrans being used as acceptor moieties, within the scope of the research conducted.” in the abstract “Line 20” does not make sense.

b2. The abstract mentioned that “Pyran groups were shown to be more capable of reducing the stated dye energy gap …”,

b3. Line 40: “Were produced” 

b4. Line 49: “to produce small molecules”

b5. Line 92 “subjected to”

b6. Line 93, “absorption spectra induced by the Pyran components”, does it mean the absorption is not from the whole molecule but the Pyran components?

b7. I suggest one or more paragraph breaks in those positions for better readability, Line 70 after [27-29]; Line 81 after “as acceptor component”; Line 89 after [44-47]. 

b8. SInce the fill factor ff, and the incident light intensity p is not discussed, I suggest to rewrite Eq. 1 as \eta \propto J_SC * V_OC. 

c.) Minor comments, typos:

c1. Please be consistent in the use of symbols. Line 100, the energy gap is “Eg” and in Table 1 as “Eg”, italic.

c2. Line 113 “recoded”, is it “recorded”?

c3. Please avoid the page break for Table 2 if possible.

c4. Inconsistent symbols. Line 204 “OC” v.s. Line 205 “OC”. Also it appears on Line 214. Same for “SC”.

c5. It seems to me that unintended indents appear on Lines 205, 212, 221, 224.

c6. Mixed use of italic and normal fonts for super/subscripts. For example, in Table 3 the superscript is in the normal fonts, while in Eq. 3, 4, 5, 6 are in italics. Also, there is also inconsistency, like ff in Line 204, but FF in Line 205. Similarly for p, \Delta G^{inject} (Line 239 v.s. Line 237).  The inconsistency appears significantly in the paragraph starting at Line 202.

c7. Please avoid the page break for Table 4 as well.

Comments on the Quality of English Language

I wanted to add a note about my choice in the option "Quality of English Language”. I think the wording and grammar of the article are fine, but the authors may benefit from professional editing services to improve the organization and flow of the whole article.

Author Response

Response to Reviewer #1's comments

(Manuscript: molecules-3176359)

We are very much thankful to the reviewer for his deep and thorough review. We have revised our present research paper in light of his useful suggestions and comments. We hope our revision has improved the paper to a level of satisfaction. Number-wise answers to his specific comments and suggestions are as follows:

Comment 1;

The results and conclusions are not convincing. There is much to discuss regarding the computational protocols that are employed in the study (see comment [4] below). Even if we assume the protocol is fine for now, a large missing piece is how those results can guide the design of the DSSC experiments. Only one experiment data point is presented throughout the article to validate the theoretical results, and in my opinion, it is far from enough. Moreover, in the energy level results, BC (cyanoacrylic group) still has the best EA, and the Eg and IP difference between BC-Pyran2 and BC could be under theoretical uncertainty (given that not enough benchmarks are provided in the article). It is not the case that a small “recorded high” energy value guarantees a successful design of the solar cells. The authors need to bridge the gap between the evaluated quantities and the realistic cell performance.

 Another example is on Line 125, which reads that “Intramolecular charge transfer (ICT) within a molecule can be determined by analysis of the electron density distribution”. What is the degree of “determined” and what can be “determined”? The rate of charge transfer would certainly be a dynamical process and the accurate determination of it would highly likely involve statistical sampling of the reorganization energy from Boltzmann-distributed geometries; molecular dynamics simulation may be needed to correct for the dynamical recrossing factor and to devise a reaction mechanism if needed. The electron transfer mechanism could be different when different dyes are involved. A frontier molecular orbital analysis does not provide such information. The authors make strong claims but fail to provide evidence to support the claims.

 The scope of the presented study is also narrow. It focuses on DSSC, then one specific realization of DSSC with TiO2 based architecture, and it tested 3 possible D-\pi-A dyes associated with a very specific functional group as an anchoring group. The whole article is presented in a very particular context. For readers of Molecules who are not necessarily an expert on DSSCs, the article needs to provide a higher-level scope to convince them the work has enough novelty and impact. For example, the authors could consider the following questions to answer: What about the other Pyran-based structures that are not included in the 3 structures included in the study? Do the results in the study suggest a systematic way to design the acceptor moieties, or let’s say, Pyran-based moieties for the DSSC? If we look at the currently designed dyes, they are not considerably better than the previous ones and the results may only have limited significance to the community. 

Response; 

The study is a theoretical concept that offers insights for improving the performance of DSSCs, similar to many other studies that offer recommendations for theoretical vision.  Although the reviewer is right that there is no clear direction on how those results can guide the design of the DSSC experiments, the study was provided with the intention of providing motivation for experimental experts to study the effect of pyrans in A site in D-π-A. As theoreticians, we lack a lot of experimental tools to bridge the gap between the evaluated quantities and the actual performance of the cell, we are limited to introducing our proposals as theoretical proposals to members of any experimental team who can adopt them. About Line 125, which reads that “Intramolecular charge transfer (ICT) within a molecule can be determined by analysis of the electron density distribution, we believe that the reviewer interpreted the word literally,  considering the HOMO and LUMO distributions on molecule surfaces, we may determine whether or not ICT is easy,  the required proper distribution for easy ICT is satisfied if the HOMO is distributed over donor parts and the LUMO is distributed over acceptor parts; different distributions could make ICT more difficult. The authors did a thorough search but were unable to locate any work of Pyrans being tested at A site in DSSCs, then the only three structures that were studied could serve as a starting point for utilizing more Pyrans structures at A site. The results of the study did not suggest a systematic way to design the acceptor moieties; the framework of the study focused on introducing the Pyrans as acceptor moieties.

Comment 2;

2.1 The article would benefit significantly if a paragraph to describe the reactive process (maybe also including the adsorption process on the surface) could be provided before the Results section. It helps to build the connection that why would design a dye structure where “[the orbitals] has a role in promoting ICT and electron injection, which in turn leads to improved orbital overlap with the TiO2 conduction band”, “[the dye] has the greatest potential to be adsorbed on the surface of TiO2”, or “[these dyes] exhibit the most potential change in the excitation process”. Only with a description of the chemical process, the readers understand what quantities are important in the design and those calculations become meaningful.

Response;

The authors agree with the reviewer in his point, and the authors did not overlook this point; instead, they relied on the fact that the chemical process underlying the calculated quantities is a familiar fundamental for any researcher working in this field.

2.2 How is the absorption of Pyran-substituted compounds calculated in the work (Figure 4)? The article states that “The maximum absorption wavelength is then determined using the TD-DFT”, and it reads to me that a stick spectrum (i.e. peak position and intensity) can be obtained with TD-DFT calculations. Then how is the line shape and peak width obtained in this study? Is a certain dynamics model for the nuclei assumed in the calculation to predict the absorption spectrum? For example, assuming a DHO model and for the size of the molecules considered here, those calculations are affordable and can provide much insight. Note that as the authors claimed, the peak positions and the broadness are crucial to DSSC performance. The computational details provided are not sufficient enough to understand the results. 

Response;

It is a given that computational chemistry can calculate the maximum absorption wavelength (i.e., peak position and intensity) using the TD-DFT. UV absorption bands can be calculated using a vast array of computational software packages, and the line shape and peak width can be obtained using an equally vast array of visualization software programs. Here is a recent paper that discusses this topic for further clarification;

https://pubs.rsc.org/en/content/articlelanding/2015/ra/c5ra00249d/unauth

2.3 On the computational protocol, Line 324, what’s the reason that “6-311G**” is the standard basis? Have the authors investigated the effects when other basis sets are employed (e.g. double-zeta basis from def family?) “Several functionals” - what are those functionals? Can authors report the benchmark results?  Line 328, what are the “test ones”? Can authors also report the benchmark results? Fig.S1 is among the examples I am looking for, and can the authors provide more experimental quantity validation to show that this density functional captures the excitation behavior correctly, rather than a lucky error cancellation? It looks to me that the authors indeed benchmark their calculations against available experimental values for BC base compound, then use the same protocol for the prediction for Pyran 1, 2, 3 compounds. The content, if stated more clearly, will strengthen the validity of the simulation results. 

Response;

Line 328, “test ones,” refers to the several functionals that were used in the initial calculations to select the most appropriate one. This was included in the supplementary Table S1 that was uploaded with the submitted files; it may be that the editorial team forgot to send it to the reviewers. The reason that “6-311G**” was used as the standard basis in the current study is that it is the most popular basis set and is frequently regarded as the best compromise between rapidity and accuracy. Four different functionals were employed in the calculations; checking other basis sets would necessitate more computational calculations, which would not be achievable because of lengthy runs and problems with processes crashing.

2.4 The protocols of adsorption calculation are ambiguous. Line 334 states that it is a Monte Carlo approach with the simulated annealing procedure, but why “during the molecular dynamics”? Is it a Molecular Dynamics simulation? Those two are completely different simulation protocols. Also the word “MD” appears in the caption of Table 4. Does the current calculation involve time-dependent trajectories? Can authors report those? 

The quantities that are reported in Table 4 could also be described more clearly. Line 258 states that the adsorption energy “represents the energy necessary for the relaxed adsorbed components to be adsorbed onto the substrate's surface”. Does it mean the energy of the geometries where the dye molecule is located at the optimal location on the TiO2(110) surface? Is the surface/lattice relaxation considered here or not? If not, how good is the frozen surface approximation for similar systems in the literature results or benchmark studies?

 In addition, how is the value dE_ad/dN_i evaluated? Line 269 reads “dE_ad/dNi provides the energies of the adsorbate molecules after removing one component of the adsorbate.” What is the “one component of the adsorbate” to remove? I originally thought that the total adsorption (or desorption, since the thermodynamical cycle can be reversed) energy is the sum of adsorption energy plus the deformation energy but it does not match the numerical results. Can authors explain what’s their definition of dE_ad/dN_i? The authors also presented dE_ad/dN_i for DCM. (By the way, the units for the last two columns are missing in Table 5.) Does it mean the desorption of a DCM molecule on the TiO2(110) surface? Or does it mean the desorption energy of the dye molecule off the DCM bulk? The latter is not an adsorption process. The authors also need to clarify to avoid confusion among potential readers.

Response;

There are no different simulation protocols; instead, the hybrid molecular dynamic (MD) algorithms employ Monte Carlo (MC) simulation to provide an ensemble of representative configurations for a complicated macromolecular system under particular thermodynamic conditions. Specifically, "simulated annealing" in molecular dynamics refers to controlling the heating and cooling of the system to quickly locate favorable minima and get beyond energy barriers; this should be done repeatedly throughout the simulation. This is the protocol employed in the Materials Studio by the Adsorption Locator module.

https://www.researchgate.net/publication/263126729_Monte_Carlo_methods_in_Materials_Studio

https://www.ncbi.nlm.nih.gov/pmc/articles/PMC4345249/#:~:text=The%20objective%20of%20a%20Monte,the%20system%20generates%20these%20configurations.

Time-dependent trajectories are not included in current calculations since they are not one of the output outcomes from Materials Studio's Adsorption Locator module. The authors were unable to report them as a result.

Line 258, the adsorption energy “represents the energy necessary for the relaxed adsorbed components to be adsorbed onto the substrate's surface." It is a well-known and established definition of adsorption energy that could be found simply in related topics. The author believes that this is the energy required for the dye to bind to the surface of the substrate. The surface/lattice relaxation has been taken into consideration throughout the adsorption calculations. The value of dE_ad/dN_i is one of the primary outputs of the Adsorption Locator module in the Materials Studio package. The authors did not define dE_ad/dN_i energy; rather, it is one of the fundamental premises found in related topics. It is a description established by experts in this field. The dye and DCM molecules were allowed to relax across the simulated box. By nature, the dye adsorbed on the TiO2(110) surface and the light DCM molecules naturally relaxed throughout the entire box. This is how molecular dynamics operates and may be verified by related topics. dE_ad/dN_i for DCM does not means the desorption of a DCM molecule on the TiO2(110) surface nor the desorption energy of the dye molecule off the DCM bulk. The lower negative values of dE_ad/dN_i for DCM suggest that the one DCM molecule will be more difficult to absorb on the metal surface and that it will be easier to remove from the system.

2.5 The chemical structure of the considered dyes (Figure 6 and associated texts) is presented AFTER the whole section of results. In my opinion, it creates unnecessary difficulty for the readers to understand the Results.

 Line 292 to 307 provide fundamental and very helpful background knowledge (!) of the whole study. Moving this part in front of the Results section would help explain the motivation and goal of the functional group substitution in this study. Also, Reference [57] is a previous work that cannot be skipped and I suggest it be presented to the reader in the introduction given its relevance.

Response;

As suggested by the reviewer, the chemical structure of the considered dyes (Figure 6 and associated texts), as well as Lines 292 to 307 and reference [57], have been moved in front of the Results section under a new section named Chemical Design.

2.6 Line 136, where the TiO2 first appears in the Results section (“had a greater ability to induce the adsorption on the TiO2 substrate”, and note it also does not appear in the Introduction), is quite abrupt. The concept needs to be introduced first before a sudden mention of TiO2 conduction band energy.

Response;

Based on the reviewer's advice, a brief description of the TiO2 oxide material has been added to the introduction.

Comment 3;

3.1 Instead of “Phenol, phosphonate, rhodanine-3-acetic acid, and many other anchoring moieties have all been studied both theoretically and experimentally” (Line 70), can authors summarize the performance of those anchoring groups (those not Pyran or cyanoacrylic acid)? 

Response;

The reviewer's suggestion is very much appreciated by the authors. However, we are afraid that there is not sufficient time to summarize the performance of those anchoring groups during the specific time of revision, but it will be taken into consideration in future work.

3.2 On Line 80, it reads that “Within the limits of the research that has been done, Pyran compounds were very rarely selected to be used as acceptor components”, does the word “rarely selected” suggest that there exists, and not too many, previous work that considers the Pyran as the anchoring group? The authors will need to clarify and precisely state the status of literature, in particular since it is a major component of the novelty of the article.

Response;

The word “rarely selected” in the sentence (“Within the limits of the research that has been done, Pyran compounds were very rarely selected to be used as acceptor components” means there are not too many studies that specifically address Pyrans as acceptor groups, even that was not explicitly stated, such as the functionalization of the Pyran core with other electron acceptor groups.

https://www.sciencedirect.com/science/article/abs/pii/S0143720819323538

3.3 On Line 88, “DFT and TD-DFT played a major role in the design, analysis, and development of organic dyes for use in solar cell applications.” Please implement citations and summarize the the status of literature studies. Reflecting “major role”, is DFT/TDDFT sufficient to capture the electron dynamical and static correlation in the context of DSSC design? Reflecting on “design, analysis, and development”, what properties did previous literature calculate with DFT/TDDFT for the design and analysis of DSSC performance? Especially, if this article calculates a property that was underestimated but indeed is crucial in DSSC design, this would greatly strengthen the novelty of the article. A similar argument can be made reflecting “impressive results” on Line 81? What are those impressive results? 

Response;

The sentence at line 88 and the sentence that follows it include four citations. The properties that have been calculated in previous literature with DFT/TDDFT for the design and analysis of DSSC performance are included in the cited works. Any argument about those properties or about “impressive results” on Line 81 could not be accomplished due to the specific shot time of revision.

3.4 On Line 111, the reference [48] seems relevant to the DSSC design but is not mentioned in the literature review in the Introduction.

Response;

Indeed, the reference [48] seems relevant to the DSSCs, but it also discussed the full-electron donor-acceptor map (FEDAM) by Martinez, so the authors believe that it is more appropriate to cite it in its current position. 

3.5 Please implement citations of the “Molecular Electrostatic Potential” methods that are used in the study. (Line 156) Also, a sentence to introduce how the electrostatic potentials are defined or calculated would be useful to the readers.

Response;

As per the reviewer's recommendation, the citation of “Molecular Electrostatic Potential” and a sentence to introduce how electrostatic potentials are defined have been added.

Comment 4;

I suggest an explicit mention of the Pyran functional group in the title as it is the major focus of the DFT study in the article. Also similarly in the abstract, “Pyran functional group” can be mentioned along with “the substitution of the aldehyde group” (Line 16) since the authors do not attempt the substitution of the aldehyde group by other possible functional groups as anchor groups. A clear and straightforward statement can be something like “The current research studied the effects of substituting the aldehyde group in D-π-A dye with Pyran moieties using DFT and TD-DFT methods” (Line 344).

Response;

Based on reviewer recommendations, the Pyran group has been mentioned in the title, and the “Pyran group” has also been mentioned along with “the substitution of the aldehyde group” (Line 17).

Comment 5;

On Line 102, “Dyes with smaller energy gaps can absorb longer wavelengths of light more readily”, what does the “more readily” mean here? Can the authors be quantitative, or at least more specific? And why is the case that “which enhances the overall efficiency of DSSCs by expanding the spectrum of solar energy that can be captured by the device”? If the point the authors wanted to make is that the longer wavelength absorption is more efficient than the short wavelength, it would be helpful to supplement a simple equation to show the scaling of efficiency v.s. wavelength.

Response;

More readily means that a dye with a smaller energy gap is more reactive to any solar radiation than a dye with a larger energy gap. A dye with a near-infrared wave length provides considerable amounts of potential for solar cell harvesting, according to a fundamental rule governing DSSC features. Therefore, a dye with a longer wavelength of absorption is more efficient than a short wavelength in the performance of a solar cell device. This is an elementary, obvious concepts that can be found in any study related to the fundamentals of DSSCs, and listing the supporting evidence would take up too much space.

Comment 6;

Table 1 and Figure 2 both report the energy levels from the calculations. Moreover, the data reported there have significant overlap. I would suggest a merge of the Table 1 and Figure 2, as the two results are currently separated by a discussion of intramolecular charge transfer in the Pyran compounds

Response;

The author found difficulty in merging Table 1 and Figure 2. The authors may draw the attention of reviewers to the fact that all previous studies on the same topic follow the same procedure for illustrating this kind of data. It is believed that the presence of the figure gives a better and faster visualization of the location of the energy levels.

Comment 7;

On Line 192, can the authors explain why is the case that “The red-shifted absorption peaks and their broadness improve the ability to obtain an efficient organic sensitizer “? It is not straightforward to follow. Could the authors provide some argument in a quantitative manner?

Response;

Since this comment and number five reflect the same argument, the answers are included in response no. 5.

Comment 8;

How was the “Major Contribution” in Table 2 calculated? Does it mean the contribution of HOMO->LUMO transition to the S0->S1 transition? Please clarify. 

Response;

Yes, “Major Contribution” in Table 2 means the contribution of HOMO->LUMO transition to the S0->S1 transition.

Comment 9;

In the paragraph discussing the photovoltaic properties of the proposed dyes, multiple quantities are presented to provide a comprehensive metric to evaluate the conversion efficiency of the dyes in DSSCs. It would be helpful to present another row in Table 3, to straightforwardly rank the four dyes according to each matric. As an example, the additional row can be named “best candidate”, and the data can be “BC” for column E_ex, “Pyran2 for LHE”, and other columns of the same.

Response;

As suggested by the reviewer, another row has been added in Table 3 named “best candidate”, to rank the four dyes according to each matrix. 

Presentation Comments

b1. Based on reviewer recommendations, The sentence “An uncommon instance of Pyrans being used as acceptor moieties, within the scope of the research conducted.” has been removed.

b2. The sentence “Pyran groups were shown to be more capable of reducing the stated dye energy gap …”, has been edited as suggested by the reviewer.

b3. The sentence included “Were produced” has been edited as suggested by the reviewer.

b4. The sentence included “to produce small molecules” has been edited as suggested by the reviewer.

b5. The sentence included “subjected to” has been edited as suggested by the reviewer.

b6. In Line 93, “absorption spectra induced by the Pyran components” it means the absorption is from the whole molecule, including the Pyran components.

b7. More paragraph breaks in the positions suggested by the reviewer have been added.

b8. Equation 1 has been modified as recommended by the reviewer.

c.) Minor comments, typos:

c1. The energy gap symbol has been unified throughout the text.

c2. In Line 113, the word “recoded” has been corrected to “recorded”.

c3. The page break for Table 2 has been avoided.

c4. The symbols V OC and J SC have been unified throughout the text.

c5. The unintended indents have been removed as recommended by the reviewer.

c6. All the required changes have been made. 

c7. The page break for Table 4 has been avoided.

Thanks in Advance

Reviewer 2 Report

Comments and Suggestions for Authors

1.Are the three materials designed by the authors synthetic? Or can they be prepared by existing organic synthesis methods? For example, the Suzuki reaction mentioned in the article.

2.It is interesting that the authors chose several DFT sets and compared them. May I ask if London dispersion was added to be more calibrated.

3.The author mentions IR or UV diagrams that were actually tested, please provide the plots to increase the readability of the article. such as ref:Tuning the Photophysical Properties of Acceptor–Donor–Acceptor Di-2-(2-oxindolin-3-ylidene) Malononitrile Materials via Extended π–Conjugation: A Joint Experimental and Theoretical Study.

4.The study of organic dyes or organic conjugation materials is interesting and some of the references are recommended for citation.(a).Preparation of Dye Semiconductors via Coupling Polymerization Catalyzed by Two Catalysts and Application to Transistor;(b).Preparation of Novel Organic Polymer Semiconductor and its Properties in Transistors through Collaborative Theoretical and Experimental Approaches.

Author Response

Response to Reviewer #2's comments

(Manuscript: molecules-3176359)

We are very much thankful to the reviewer for his deep and thorough review. We have revised our present research paper in light of his useful suggestions and comments. We hope our revision has improved the paper to a level of satisfaction. Number-wise answers to his specific comments and suggestions are as follows:

Comment 1;

Are the three materials designed by the authors synthetic? Or can they be prepared by existing organic synthesis methods? For example, the Suzuki reaction mentioned in the article.

Response; 

The three materials that the authors suggested were not synthesized; instead, they were designed and studied entirely theoretically. However, the authors believe that the three proposed compounds could be synthesized by following the cited experimental source that provided the process for the base compound synthesis.

Comment 2;

It is interesting that the authors chose several DFT sets and compared them. May I ask if London dispersion was added to be more calibrated.

Response;

The authors would like to thank the reviewer for his suggestion. The London dispersion would be better calibrated, but we are afraid that this is not possible to accomplish because the computational facilities cannot support more computations (long time and process crashing issues). We really hope you will understand.

Comment 3;

The author mentions IR or UV diagrams that were actually tested, please provide the plots to increase the readability of the article. such as ref: Tuning the Photophysical Properties of Acceptor–Donor–Acceptor Di-2-(2-oxindolin-3-ylidene) Malononitrile Materials via Extended π–Conjugation: A Joint Experimental and Theoretical Study.

Response;

Based on the reviewer's advice, the plot for the UV diagram that was actually tested was provided as a supplementary Figure S1, and the mentioned reference was cited.

Comment 4;

The study of organic dyes or organic conjugation materials is interesting and some of the references are recommended for citation. (a). Preparation of Dye Semiconductors via Coupling Polymerization Catalyzed by Two Catalysts and Application to Transistor;(b). Preparation of Novel Organic Polymer Semiconductor and its Properties in Transistors through Collaborative Theoretical and Experimental Approaches.

Response;

As per the reviewer's recommendation, the suggested references were cited.

Thanks in Advance

Round 2

Reviewer 1 Report

Comments and Suggestions for Authors

Thank you to the authors for the prompt response. I appreciate the editions and agree that the revision has improved the quality of the article. However, with the current version, I am not able to recommend the publication of the article, especially, due to the concerns associated with the following comments. Please find my comments below in blue.

========

Reply to:

Comment 1;

The results and conclusions are not convincing. There is much to discuss regarding the computational protocols that are employed in the study (see comment [4] below). Even if we assume the protocol is fine for now, a large missing piece is how those results can guide the design of the DSSC experiments. Only one experiment data point is presented throughout the article to validate the theoretical results, and in my opinion, it is far from enough. Moreover, in the energy level results, BC (cyanoacrylic group) still has the best EA, and the Eg and IP difference between BC-Pyran2 and BC could be under theoretical uncertainty (given that not enough benchmarks are provided in the article). It is not the case that a small “recorded high” energy value guarantees a successful design of the solar cells. The authors need to bridge the gap between the evaluated quantities and the realistic cell performance.

 Another example is on Line 125, which reads that “Intramolecular charge transfer (ICT) within a molecule can be determined by analysis of the electron density distribution”. What is the degree of “determined” and what can be “determined”? The rate of charge transfer would certainly be a dynamical process and the accurate determination of it would highly likely involve statistical sampling of the reorganization energy from Boltzmann-distributed geometries; molecular dynamics simulation may be needed to correct for the dynamical recrossing factor and to devise a reaction mechanism if needed. The electron transfer mechanism could be different when different dyes are involved. A frontier molecular orbital analysis does not provide such information. The authors make strong claims but fail to provide evidence to support the claims.

 The scope of the presented study is also narrow. It focuses on DSSC, then one specific realization of DSSC with TiO2 based architecture, and it tested 3 possible D-\pi-A dyes associated with a very specific functional group as an anchoring group. The whole article is presented in a very particular context. For readers of Molecules who are not necessarily an expert on DSSCs, the article needs to provide a higher-level scope to convince them the work has enough novelty and impact. For example, the authors could consider the following questions to answer: What about the other Pyran-based structures that are not included in the 3 structures included in the study? Do the results in the study suggest a systematic way to design the acceptor moieties, or let’s say, Pyran-based moieties for the DSSC? If we look at the currently designed dyes, they are not considerably better than the previous ones and the results may only have limited significance to the community. 

Response; 

The study is a theoretical concept that offers insights for improving the performance of DSSCs, similar to many other studies that offer recommendations for theoretical vision.  Although the reviewer is right that there is no clear direction on how those results can guide the design of the DSSC experiments, the study was provided with the intention of providing motivation for experimental experts to study the effect of pyrans in A site in D-π-A. As theoreticians, we lack a lot of experimental tools to bridge the gap between the evaluated quantities and the actual performance of the cell, we are limited to introducing our proposals as theoretical proposals to members of any experimental team who can adopt them. 

I don’t find the response sufficiently addresses the concerns in the previous section. I understand that it is a theoretical study and I did not suggest the authors use “a lot of experimental tools“ to bridge the gap between theoretical and experimental results. 

Theoretical models need to be validated with the experimental measurements (note, not necessary from the authors themselves), otherwise, the predictions could be detached from the chemistry reality. As the response agreed, “there is no clear direction on how those results can guide the design of the DSSC experiments”. The overlook of the connection with the experiments narrows the significance and impact of a theoretical study.

Maybe I missed it, but I am not able to find the response to the following comment in the first review:

Only one experiment data point is presented throughout the article to validate the theoretical results, and in my opinion, it is far from enough. Moreover, in the energy level results, BC (cyanoacrylic group) still has the best EA, and the Eg and IP difference between BC-Pyran2 and BC could be under theoretical uncertainty (given that not enough benchmarks are provided in the article). It is not the case that a small “recorded high” energy value guarantees a successful design of the solar cells. The authors need to bridge the gap between the evaluated quantities and the realistic cell performance.

About Line 125, which reads that “Intramolecular charge transfer (ICT) within a molecule can be determined by analysis of the electron density distribution, we believe that the reviewer interpreted the word literally,  considering the HOMO and LUMO distributions on molecule surfaces, we may determine whether or not ICT is easy,  the required proper distribution for easy ICT is satisfied if the HOMO is distributed over donor parts and the LUMO is distributed over acceptor parts; different distributions could make ICT more difficult. The authors did a thorough search but were unable to locate any work of Pyrans being tested at A site in DSSCs, then the only three structures that were studied could serve as a starting point for utilizing more Pyrans structures at A site. The results of the study did not suggest a systematic way to design the acceptor moieties; the framework of the study focused on introducing the Pyrans as acceptor moieties.

I suggest adding the understanding in this response to the article.

========

Reply to:

2.2 How is the absorption of Pyran-substituted compounds calculated in the work (Figure 4)? The article states that “The maximum absorption wavelength is then determined using the TD-DFT”, and it reads to me that a stick spectrum (i.e. peak position and intensity) can be obtained with TD-DFT calculations. Then how is the line shape and peak width obtained in this study? Is a certain dynamics model for the nuclei assumed in the calculation to predict the absorption spectrum? For example, assuming a DHO model and for the size of the molecules considered here, those calculations are affordable and can provide much insight. Note that as the authors claimed, the peak positions and the broadness are crucial to DSSC performance. The computational details provided are not sufficient enough to understand the results.

Response;

It is a given that computational chemistry can calculate the maximum absorption wavelength (i.e., peak position and intensity) using the TD-DFT. UV absorption bands can be calculated using a vast array of computational software packages, and the line shape and peak width can be obtained using an equally vast array of visualization software programs. Here is a recent paper that discusses this topic for further clarification;

https://pubs.rsc.org/en/content/articlelanding/2015/ra/c5ra00249d/unauth

It seems that the authors avoid answering directly to “how is the line shape and peak width obtained in this study” in their response. As shown in my earlier comment, I understand that “computational chemistry can calculate the maximum absorption wavelength (i.e., peak position and intensity) using the TD-DFT”. The information I am seeking is on the peak width and line shape. Those quantities are also results from a computational chemistry calculation, and are not supposed to be “obtained using an equally vast array of visualization software programs”. 

I understand that the authors rely on computational software packages to obtain the results, however, I do believe it is the author’s responsibility to understand the software tool they are using and to provide the readers with an accurate impression of the computation methods they employed.

I read through the article the authors provided but was not able to find the needed information. Please identify the specific section/page.

========

Reply to:

Comment 5;

On Line 102, “Dyes with smaller energy gaps can absorb longer wavelengths of light more readily”, what does the “more readily” mean here? Can the authors be quantitative, or at least more specific? And why is the case that “which enhances the overall efficiency of DSSCs by expanding the spectrum of solar energy that can be captured by the device”? If the point the authors wanted to make is that the longer wavelength absorption is more efficient than the short wavelength, it would be helpful to supplement a simple equation to show the scaling of efficiency v.s. wavelength.

Response;

More readily means that a dye with a smaller energy gap is more reactive to any solar radiation than a dye with a larger energy gap. A dye with a near-infrared wave length provides considerable amounts of potential for solar cell harvesting, according to a fundamental rule governing DSSC features. Therefore, a dye with a longer wavelength of absorption is more efficient than a short wavelength in the performance of a solar cell device. This is an elementary, obvious concepts that can be found in any study related to the fundamentals of DSSCs, and listing the supporting evidence would take up too much space.

In my opinion, the argument in the response lacks scientific preciseness. The authors stated that “More readily means that a dye with a smaller energy gap is more reactive to any solar radiation than a dye with a larger energy gap”. “Reactive” means chemical bond break and formation while “solar radiation” is a photon absorption process. Moreover, it is not “any” solar radiation, because as Fermi’s golden rule stated, the material absorbs the light with its wavelength in close resonance with the electronic energy gap. 

It is also a basic physical chemistry concept, and the authors need to clarify if it is not what they trying to express. For example, in the lecture notes by Dr. Jim Clark, (https://chem.libretexts.org/Courses/Montana_State_University/MSU%3A_CHMY_362_Elements_of_Physical_Chemistry/06%3A_UV-Vis/6.01%3A_What_Causes_Molecules_to_Absorb_UV_and_Visible_Light)

“In each possible case, an electron is excited from a full orbital into an empty anti-bonding orbital. Each jump takes energy from the light, and a big jump obviously needs more energy than a small one. Each wavelength of light has a particular energy associated with it. If that particular amount of energy is just right for making one of these energy jumps, then that wavelength will be absorbed - its energy will have been used in promoting an electron.”

========

Reply to:

Comment 2;

2.1 The article would benefit significantly if a paragraph to describe the reactive process (maybe also including the adsorption process on the surface) could be provided before the Results section. It helps to build the connection that why would design a dye structure where “[the orbitals] has a role in promoting ICT and electron injection, which in turn leads to improved orbital overlap with the TiO2 conduction band”, “[the dye] has the greatest potential to be adsorbed on the surface of TiO2”, or “[these dyes] exhibit the most potential change in the excitation process”. Only with a description of the chemical process, the readers understand what quantities are important in the design and those calculations become meaningful.

Response;

The authors agree with the reviewer in his point, and the authors did not overlook this point; instead, they relied on the fact that the chemical process underlying the calculated quantities is a familiar fundamental for any researcher working in this field.

I am not quite convinced by this response. I believe that it is not the authors’ intention to exclude the researcher who does not work in the field of DSSCs as their readers.

========

Reply to:

Comment 3;

3.1 Instead of “Phenol, phosphonate, rhodanine-3-acetic acid, and many other anchoring moieties have all been studied both theoretically and experimentally” (Line 70), can authors summarize the performance of those anchoring groups (those not Pyran or cyanoacrylic acid)? 

Response;

The reviewer's suggestion is very much appreciated by the authors. However, we are afraid that there is not sufficient time to summarize the performance of those anchoring groups during the specific time of revision, but it will be taken into consideration in future work.

The lack of a suggested summary significantly undermines the validity of the conclusion. The authors emphasized e.g. in the abstract, that “Pyran groups reduced the stated dye energy gap better than other known anchoring groups” (Line 21). This conclusion relies on knowledge of the performance of “other known anchoring groups and therefore including the summary is necessary and cannot be excused based on the time limit.

In this case, I would suggest the authors ask for a time extension from the Editorial office of the journal. To clarify, I did not suggest the collection of the new data.

========

Reply to:

3.3 On Line 88, “DFT and TD-DFT played a major role in the design, analysis, and development of organic dyes for use in solar cell applications.” Please implement citations and summarize the the status of literature studies. Reflecting “major role”, is DFT/TDDFT sufficient to capture the electron dynamical and static correlation in the context of DSSC design? Reflecting on “design, analysis, and development”, what properties did previous literature calculate with DFT/TDDFT for the design and analysis of DSSC performance? Especially, if this article calculates a property that was underestimated but indeed is crucial in DSSC design, this would greatly strengthen the novelty of the article. A similar argument can be made reflecting “impressive results” on Line 81? What are those impressive results? 

Response;

The sentence at line 88 and the sentence that follows it include four citations. The properties that have been calculated in previous literature with DFT/TDDFT for the design and analysis of DSSC performance are included in the cited works. Any argument about those properties or about “impressive results” on Line 81 could not be accomplished due to the specific shot time of revision.

Similar to the above comments. If the authors agree that more work needs to be done to improve the scientific soundness and the impact of the article to meet the standard of publications, I support them in asking for a time extension from the Editorial office to complete any calculation of value.

Comments on the Quality of English Language

The article reads much better than the original version. I don't think a professional edition service is needed anymore.

Author Response

Response to Reviewer #1's comments

(Manuscript: molecules-3176359)

We are very much thankful to the reviewer for his deep and thorough review. We have revised our present research paper in light of his useful suggestions and comments. We hope our revision has improved the paper to a level of satisfaction. Number-wise answers to his specific comments and suggestions are as follows:

Comment 1;

I don’t find the response sufficiently addresses the concerns in the previous section. I understand that it is a theoretical study and I did not suggest the authors use “a lot of experimental tools “to bridge the gap between theoretical and experimental results.

Theoretical models need to be validated with the experimental measurements (note, not necessary from the authors themselves), otherwise, the predictions could be detached from the chemistry reality. As the response agreed, “there is no clear direction on how those results can guide the design of the DSSC experiments”. The overlook of the connection with the experiments narrows the significance and impact of a theoretical study.

Maybe I missed it, but I am not able to find the response to the following comment in the first review:

I suggest adding the understanding in this response to the article.

 Response; 

There have been numerous previous DFT and ab initio studies that did not include the type of required validation with the experimental measurements. Our previous work (The Effect of Conjugated Nitrile Structures as Acceptor Moieties on the Photovoltaic Properties of Dye-Sensitized Solar Cells: DFT and TD-DFT Investigation)-pointed by the reviewer-that has been published in the International Journal of Molecular Sciences, one of MDPI journals, does not contain the type of validation asked by the reviewer, and also more powerful theoretical studies published in reputable, well-known journals. Therefore, we hope that the reviewer will regard this work as a theoretical proposal in an effort to attract interest in the kind of materials that have been designed.

About Line 125, the reviewer comment is “I suggest adding the understanding in this response to the article”; As suggested by the reviewer, the understanding in this response was added to the article.

The reviewer said "Moreover, in the energy level results, BC (cyanoacrylic group) still has the best EA, and the Eg and IP difference between BC-Pyran2 and BC could be under theoretical uncertainty". It prompts the question of why does the EA of (BC-cyanoacrylic) not fall within the same theoretical uncertainty as that of Eg and IP, as indicated by the reviewer? In addition, it is hard to disregard the small energy gap as a critical component of DSSCs.

Comment 2;

seems that the authors avoid answering directly to “how is the line shape and peak width obtained in this study” in their response. As shown in my earlier comment, I understand that “computational chemistry can calculate the maximum absorption wavelength (i.e., peak position and intensity) using the TD-DFT”. The information I am seeking is on the peak width and line shape. Those quantities are also results from a computational chemistry calculation, and are not supposed to be “obtained using an equally vast array of visualization software programs”.

I understand that the authors rely on computational software packages to obtain the results, however, I do believe it is the author’s responsibility to understand the software tool they are using and to provide the readers with an accurate impression of the computation methods they employed.

I read through the article the authors provided but was not able to find the needed information. Please identify the specific section/page.

Response;

The reviewer asked to provide an accurate impression of the computation methods they employed via his comment “how is the line shape and peak width obtained in this study” regarding the UV spectra. The answer is we do not have much details about this point. Throughout the several research projects we have previously provided, this is the first time we have come across a query like this one. Moreover, in related similar studies, we could not find details on how the line shape and peak width of UV spectra are obtained using TD-DFT calculations.

Comment 3;

In my opinion, the argument in the response lacks scientific preciseness. The authors stated that “More readily means that a dye with a smaller energy gap is more reactive to any solar radiation than a dye with a larger energy gap”. “Reactive” means chemical bond break and formation while “solar radiation” is a photon absorption process. Moreover, it is not “any” solar radiation, because as Fermi’s golden rule stated, the material absorbs the light with its wavelength in close resonance with the electronic energy gap.

It is also a basic physical chemistry concept, and the authors need to clarify if it is not what they trying to express. For example, in the lecture notes by Dr. Jim Clark, (https://chem.libretexts.org/Courses/Montana_State_University/MSU%3A_CHMY_362_Elements_of_Physical_Chemistry/06%3A_UV-Vis/6.01%3A_What_Causes_Molecules_to_Absorb_UV_and_Visible_Light)

“In each possible case, an electron is excited from a full orbital into an empty anti-bonding orbital. Each jump takes energy from the light, and a big jump obviously needs more energy than a small one. Each wavelength of light has a particular energy associated with it. If that particular amount of energy is just right for making one of these energy jumps, then that wavelength will be absorbed - its energy will have been used in promoting an electron.”.

Response;

It seems like the word was interpreted literally. "More reactive to any solar radiation." Reactive here means more ability to absorb solar radiation.

Comment 4;

I am not quite convinced by this response. I believe that it is not the authors’ intention to exclude the researcher who does not work in the field of DSSCs as their readers.

Response;

There are no details that could be introduced regarding the "reactive process.". Once again, this is considered familiar and fundamental for any researcher working in this field. For example, introducing a description of the reactive process for the adsorption process on the surface as asked by the reviewer would be padding that is useless in research papers since everyone in the field is aware of these basics. 

Comment 5;

The lack of a suggested summary significantly undermines the validity of the conclusion. The authors emphasized e.g. in the abstract, that “Pyran groups reduced the stated dye energy gap better than other known anchoring groups” (Line 21). This conclusion relies on knowledge of the performance of “other known anchoring groups and therefore including the summary is necessary and cannot be excused based on the time limit.

In this case, I would suggest the authors ask for a time extension from the Editorial office of the journal. To clarify, I did not suggest the collection of the new data.

Response;

The referred theoretically and experimentally studies of Phenol, phosphate, rhodanine-3-acetic acid, and many other anchoring moieties have all been done on organic dyes with totally different structures from our D-π-Aldehyde Dye. Therefore, the required summary won't prove to be informative.

Comment 6;

Similar to the above comments. If the authors agree that more work needs to be done to improve the scientific soundness and the impact of the article to meet the standard of publications, I support them in asking for a time extension from the Editorial office to complete any calculation of value.

Response;

The same response as in the previous comment, the sentence "DFT and TD-DFT played a major role in the design, analysis, and development of organic dyes for use in solar cell applications" was mentioned generally as a literature citation, and their included dyes are totally different structures than our D-π-Aldehyde Dye. 

 Thanks in Advance
